# SVGD as a kernelized Wasserstein gradient flow of the chi-squared divergence

**Sinho Chewi**
MIT
schewi@mit.edu

**Thibaut Le Gouic**
MIT
tlegouic@mit.edu

**Chen Lu**
MIT
chenl819@mit.edu

**Tyler Maunu**
MIT
maunut@mit.edu

**Philippe Rigollet**
MIT
rigollet@mit.edu

## Abstract

Stein Variational Gradient Descent (SVGD), a popular sampling algorithm, is often described as the kernelized gradient flow for the Kullback-Leibler divergence in the geometry of optimal transport. We introduce a new perspective on SVGD that instead views SVGD as the (kernelized) gradient flow of the chi-squared divergence which, we show, exhibits a strong form of uniform exponential ergodicity under conditions as weak as a Poincaré inequality. This perspective leads us to propose an alternative to SVGD, called Laplacian Adjusted Wasserstein Gradient Descent (LAWGD), that can be implemented from the spectral decomposition of the Laplacian operator associated with the target density. We show that LAWGD exhibits strong convergence guarantees and good practical performance.

## 1 Introduction

The seminal paper of Jordan, Kinderlehrer, and Otto [JKO98] has profoundly reshaped our understanding of sampling algorithms. What is now commonly known as the *JKO scheme* interprets the evolution of marginal distributions of a Langevin diffusion as a gradient flow of a Kullback-Leibler (KL) divergence over the Wasserstein space of probability measures. This optimization perspective on Markov Chain Monte Carlo (MCMC) has not only renewed our understanding of algorithms based on Langevin diffusions [Dal17a; Ber18; CB18; Wib18; DMM19; VW19], but has also fueled the discovery of new MCMC algorithms inspired by the diverse and powerful optimization toolbox [Mar+12; Sim+16; Che+18; Ber18; Hsi+18; Wib18; Ma+19; Wib19; Che+20; DR20; Zha+20b].

In order to arrive at a practical sampling algorithm, one must discretize the Wasserstein gradient flow. The most common discretization is to discretize the Langevin diffusion, resulting in the Unadjusted Langevin Algorithm (ULA) [Dal17b; DM17]. However, it is unclear whether this diffusion based discretization is the most effective one. In fact, ULA is asymptotically biased, which results in slow convergence and often requires ad-hoc adjustments [Dwi+19]. To overcome this limitation, various methods that track the Wasserstein gradient flow more closely have been recently developed [Ber18; Wib18; SKL20].

An alternative sampling approach that avoids diffusions is to construct a sequence of *deterministic* mappings that approximately pushes forward an initial distribution to the target distribution. Let $F$ denote a functional over the Wasserstein space of distributions. The Wasserstein gradient flow of $F$ may be described as the deterministic and time-inhomogeneous Markov process $(X_t)_{t \geq 0}$ started at a random variable $X_0 \sim \mu_0$ and evolving according to $\dot{X}_t = -[\nabla_{W_2} F(\mu_t)](X_t)$, where $\mu_t$ denotes the distribution of $X_t$. Here $[\nabla_{W_2} F(\mu)](\cdot) : \mathbb{R}^d \to \mathbb{R}^d$ is the Wasserstein gradient of $F$ at $\mu$. If

$F(\mu) = D_{\mathrm{KL}}(\mu \parallel \pi)$, where $\pi \propto e^{-V}$ is a given target distribution on $\mathbb{R}^d$, it is known [AGS08; Vil09; San17] that $\nabla_{W_2} F(\mu) = \nabla \ln(\mathrm{d}\mu/\mathrm{d}\pi)$. Therefore, a natural discretization of the Wasserstein gradient flow with step size $h > 0$, albeit one that cannot be implemented since it depends on the distribution $\mu_t$ of $X_t$, is:

$$X_{t+1} = X_t - h\nabla \ln\big(\frac{\mathrm{d}\mu_t}{\mathrm{d}\pi}(X_t)\big), \qquad t = 0, 1, 2, \ldots.$$

While $\mu_t$ can, in principle, be estimated by evolving a large number of particles $X_t^{[1]}, \ldots, X_t^{[N]}$, estimation of $\mu_t$ is hindered by the curse of dimensionality and this approach still faces significant computational challenges despite attempts to improve the original JKO scheme [SKL20; WL20].

A major advance in this direction was achieved by allowing for *approximate* Wasserstein gradients, which makes the push forward maps tractable. More specifically, Stein Variational Gradient Descent (SVGD), recently proposed by [LW16] (see Section 2 for more details), consists in replacing $\nabla_{W_2} F(\mu)$ by its image $\mathcal{K}_\mu \nabla_{W_2} F(\mu)$ under the integral operator $\mathcal{K}_\mu : L^2(\mu) \to L^2(\mu)$ associated to a chosen kernel $K : \mathbb{R}^d \times \mathbb{R}^d \to \mathbb{R}$ and defined by $\mathcal{K}_\mu f(x) := \int K(x, y) f(y) \, \mathrm{d}\mu(y)$ for $f \in L^2(\mu)$. This leads to the following process:

$$X_{t+1} = X_t - h[\mathcal{K}_{\mu_t} \nabla_{W_2} F(\mu_t)](X_t), \qquad t = 0, 1, 2, \ldots. \tag{SVGD$_\mathsf{p}$}$$

where we apply the integral operator $\mathcal{K}_{\mu_t}$ individually to each coordinate of the Wasserstein gradient. In turn, this *kernelization trick* overcomes most of the above computational bottleneck. Building on this perspective, [DNS19] introduced a new geometry, different from the Wasserstein geometry and which they call the *Stein geometry*, in which the continuous limit of (SVGD$_\mathsf{p}$) becomes the gradient flow of the KL divergence.

However, despite this recent advance, the theoretical properties of SVGD are still largely unexplored, resulting in little understanding of SVGD's known problems, such as mode collapse or a lack of guidance on how to choose an appropriate kernel $K$. Consequently diffusion-based algorithms remain the dominant choice for applications. In this work, we we provide a new and stronger theoretical footing for the development of such deterministic mappings.

**Our contributions**. We introduce, in Section 2.3, a new perspective on SVGD by viewing it as kernelized gradient flow of the chi-squared divergence rather than the KL divergence. This perspective is fruitful in two ways. First, it uses a single integral operator $\mathcal{K}_\pi$—as opposed to (SVGD$_\mathsf{p}$), which requires a family of integral operators $\mathcal{K}_\mu$, $\mu \ll \pi$—providing a conceptually clear guideline for choosing $K$, namely: $K$ should be chosen to make $\mathcal{K}_\pi$ approximately equal to the identity operator. Second, under the idealized choice $\mathcal{K}_\pi = \mathrm{id}$, we show that this gradient flow converges exponentially fast in KL divergence as soon as the target distribution $\pi$ satisfies a Poincaré inequality. In fact, our results are stronger than exponential convergence and they highlight *strong uniform ergodicity*: the gradient flow forgets the initial distribution after a finite time that is at most half of the Poincaré constant. To establish this exponential convergence under a relatively weak con-

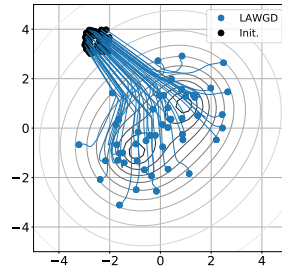

Figure 1: Sampling from a mixture of two 2D Gaussians with LAWGD. See Appendix C.

dition (Poincaré inequality), we employ the following technique. While the gradient flow aims at minimizing the chi-squared divergence by following the curve in Wasserstein space with steepest descent, we do not track its progress with the objective function itself, the chi-squared divergence, but instead we track it with the KL divergence. This is in a sense dual to argument employed in [Che+20], where the chi-squared divergence is used to track the progress of a gradient flow on the KL divergence. A more standard analysis relying on Łojasiewicz inequalities also yields rates of convergence on the chi-squared divergence under stronger assumptions such as a log-Sobolev inequality, and log-concavity. These results establish the first finite-time theoretical guarantees for SVGD in an idealized setting.

Beyond providing a better understanding of SVGD, our novel perspective is instrumental in the development of a new sampling algorithm, which we call Laplacian Adjusted Wasserstein Gradient Descent (LAWGD) and present in Section 4. We show that it possesses a striking theoretical property:

assuming that the target distribution $\pi$ satisfies a Poincaré inequality, LAWGD converges exponentially fast, with *no* dependence on the Poincaré constant. This scale invariance has been recently demonstrated for the Newton-Langevin diffusion [Che+20], but under the additional assumption that $\pi$ is log-concave. A successful implementation of LAWGD hinges on the spectral decomposition of a certain differential operator which is within reach of modern PDE solvers. As a proof of concept, we show that LAWGD, implemented using a naïve finite differences method, performs well on mixtures of Gaussians in one or two dimensions, whereas SVGD fails. This is an indication that our novel perspective could be the correct one to further advance the state-of-the-art for sampling via deterministic mappings. Implementing LAWGD in high dimensions is challenging, and we are not advocating for it as the definitive solution of the sampling problem. Instead, LAWGD serves as the start of a family of interacting particle systems with an interacting potential that depends strongly and non-trivially on the target distribution and furthermore comes with strong theoretical guarantees. We hope this work can encourage further research in the application of numerical PDEs for sampling.

**Related work**. Since its introduction in [LW16], a number of variants of SVGD have been considered. They include a stochastic version [Li+20], a version that approximates the Newton direction in Wasserstein space [Det+18], a version that uses matrix kernels [Wan+19], an accelerated version [Liu+19], and a hybrid with Langevin [Zha+20a]. Several works have studied theoretical properties of SVGD, including its interpretation as a gradient flow under a modified geometry [Liu17; DNS19], and its asymptotic convergence [LLN19].

**Notation**. In this paper, all probability measures are assumed to have densities w.r.t. Lebesgue measure; therefore, we frequently abuse notation by identifying a probability measure with its Lebesgue density. For a differentiable kernel $K : \mathbb{R}^d \times \mathbb{R}^d \to \mathbb{R}$, we denote by $\nabla_1 K : \mathbb{R}^d \times \mathbb{R}^d \to \mathbb{R}^d$ (resp. $\nabla_2 K$) the gradient of the kernel w.r.t. the first (resp. second) argument. When describing particle algorithms, we use a subscript to denote the time index and brackets to denote the particle index, i.e., $X_t^{[i]}$ refers to the $i$th particle at time (or iteration number) $t$.

## 2 SVGD as a kernelized Wasserstein gradient flow

### 2.1 Wasserstein gradient flows

In this section, we review the theory of gradient flows on the space $\mathcal{P}_{2,\mathrm{ac}}(\mathbb{R}^d)$ of probability measures absolutely continuous w.r.t. Lebesgue measure and possessing a finite second moment, equipped with the 2-Wasserstein metric $W_2$. We refer readers to [Vil03; San15; San17] for introductory treatments of optimal transport, and to [AGS08; Vil09] for detailed treatments of Wasserstein gradient flows.

Let $F : \mathcal{P}_{2,\mathrm{ac}}(\mathbb{R}^d) \to \mathbb{R} \cup \{\infty\}$ be a functional defined on Wasserstein space. We say that a curve $(\mu_t)_{t\geq 0}$ of probability measures is a *Wasserstein gradient flow* for the functional $F$ if it satisfies

$$\partial_t \mu_t = \mathrm{div}\big(\mu_t \nabla_{W_2} F(\mu_t)\big) \tag{1}$$

in a weak sense. Here, $\nabla_{W_2} F(\mu) := \nabla \delta F(\mu)$ is the Wasserstein gradient of the functional $F$ at $\mu$, where $\delta F(\mu) : \mathbb{R}^d \to \mathbb{R}$ is the *first variation* of $F$ at $\mu$, defined by

$$\lim_{\varepsilon \to 0} \frac{F(\mu + \varepsilon \xi) - F(\mu)}{\varepsilon} = \int \delta F(\mu) \, \mathrm{d}\xi, \qquad \text{for all } \xi \text{ with } \int \mathrm{d}\xi = 0,$$

and $\nabla$ denotes the usual (Euclidean) gradient. Hence, the Wasserstein gradient, at each $\mu \in \mathcal{P}_{2,\mathrm{ac}}(\mathbb{R}^d)$, is a map from $\mathbb{R}^d$ to $\mathbb{R}^d$.

Using the *continuity equation*, we can give an Eulerian interpretation to the evolution equation (1) (see [San15, §4] and [AGS08, §8]). Given a family of vector fields $(v_t)_{t\geq 0}$, let $(X_t)_{t\geq 0}$ be a curve in $\mathbb{R}^d$ with random initial point $X_0 \sim \mu_0$, and such that $(X_t)_{t\geq 0}$ is an integral curve of the vector fields $(v_t)_{t\geq 0}$, that is, $\dot{X}_t = v_t(X_t)$. If we let $\mu_t$ denote the law of $X_t$, then $(\mu_t)_{t\geq 0}$ evolves according to the *continuity equation*

$$\partial_t \mu_t = -\mathrm{div}(\mu_t v_t). \tag{2}$$

Comparing (1) and (2), we see that (1) describes the evolution of the marginal law $(\mu_t)_{t\geq 0}$ of the curve $(X_t)_{t\geq 0}$ with $X_0 \sim \mu_0$ and $\dot{X}_t = -[\nabla_{W_2} F(\mu_t)](X_t)$.

Wasserstein calculus provides the following (formal) calculation rule: the Wasserstein gradient flow $(\mu_t)_{t\geq 0}$ for the functional $F$ dissipates $F$ at the rate $\partial_t F(\mu_t) = -\mathbb{E}_{\mu_t}[\|\nabla_{W_2} F(\mu_t)\|^2]$. More generally, for a curve $(\mu_t)_{t\geq 0}$ evolving according to the continuity equation (2), the time-derivative of $F$ is given by $\partial_t F(\mu_t) = \mathbb{E}_{\mu_t}\langle \nabla_{W_2} F(\mu_t), v_t\rangle$.

In this paper, we are primarily concerned with two functionals: the Kullback-Leibler (KL) divergence $D_{\mathrm{KL}}(\cdot \,\|\, \pi)$, and the chi-squared divergence $\chi^2(\cdot \,\|\, \pi)$ (see, e.g., [Tsy09]). It is a standard exercise [AGS08; San15] to check that Wasserstein gradients of these functionals are, respectively,

$$\left(\nabla_{W_2} D_{\mathrm{KL}}(\cdot \,\|\, \pi)\right)(\mu) = \nabla \ln \frac{\mathrm{d}\mu}{\mathrm{d}\pi}, \qquad \left(\nabla_{W_2} \chi^2(\cdot \,\|\, \pi)\right)(\mu) = 2\nabla \frac{\mathrm{d}\mu}{\mathrm{d}\pi}. \tag{3}$$

## 2.2 SVGD as a kernelized gradient flow of the KL divergence

SVGD[1] is achieved by replacing the Wasserstein gradient $\nabla \ln(\mathrm{d}\mu_t/\mathrm{d}\pi)$ of the KL divergence with $\mathcal{K}_{\mu_t}\nabla \ln(\mathrm{d}\mu_t/\mathrm{d}\pi)$, leading to the particle evolution equation ($\mathrm{SVGD_p}$).

Recalling that $\pi \propto e^{-V}$, we get

$$\mathcal{K}_{\mu_t}\nabla \ln \frac{\mathrm{d}\mu_t}{\mathrm{d}\pi}(x) := \int K(x,\cdot)\nabla \ln \frac{\mathrm{d}\mu_t}{\mathrm{d}\pi}\,\mathrm{d}\mu_t = \int K(x,\cdot)\nabla V\,\mathrm{d}\mu_t - \int \nabla_2 K(x,\cdot)\,\mathrm{d}\mu_t, \tag{4}$$

where, in the second identity, we used integration by parts. This expression shows that rather than having to estimate the distribution $\mu_t$, it is sufficient to estimate the expectation $\int \nabla_2 K(x,\cdot)\,\mathrm{d}\mu_t$. This is the key to the computational tractability of SVGD. Indeed, the kernelized gradient flow can implemented by drawing $N$ particles $X_0^{[1]}, \ldots, X_0^{[N]} \overset{\text{i.i.d.}}{\sim} \mu_0$ and following the coupled dynamics

$$\dot{X}_t^{[i]} = -\mathcal{K}_{\mu_t}\nabla \ln \frac{\mathrm{d}\mu_t}{\mathrm{d}\pi}(X_t^{[i]}) = -\int K(X_t^{[i]},\cdot)\nabla V\,\mathrm{d}\mu_t + \int \nabla_2 K(X_t^{[i]},\cdot)\,\mathrm{d}\mu_t, \qquad i \in [N].$$

With this, we can simply estimate the expectation with respect to $\mu_t$ with an average over all particles, which yeilds the SVGD algorithm:

$$X_{t+1}^{[i]} = X_t^{[i]} - \frac{h}{N}\sum_{j=1}^{N} K(X_t^{[i]}, X_t^{[j]})\nabla V(X_t^{[j]}) + \frac{h}{N}\sum_{j=1}^{N} \nabla_2 K(X_t^{[i]}, X_t^{[j]}), \qquad i \in [N]. \tag{5}$$

## 2.3 SVGD as a kernelized gradient flow of the chi-squared divergence

Recall from Section 2.1 that by the continuity equation, the continuous limit of the particle evolution equation ($\mathrm{SVGD_p}$) translates into the following PDE that describes the evolution of the distribution $\mu_t$ of $X_t$:

$$\partial_t \mu_t = \mathrm{div}\left(\mu_t \mathcal{K}_{\mu_t}\nabla \ln \frac{\mathrm{d}\mu_t}{\mathrm{d}\pi}\right). \tag{$\mathrm{SVGD_d}$}$$

We make the simple observation that

$$\mathcal{K}_{\mu_t}\nabla \ln \frac{\mathrm{d}\mu_t}{\mathrm{d}\pi}(x) = \int K(x,y)\nabla \ln \frac{\mathrm{d}\mu_t}{\mathrm{d}\pi}(y)\,\mathrm{d}\mu_t(y) = \int K(x,y)\nabla \frac{\mathrm{d}\mu_t}{\mathrm{d}\pi}(y)\,\mathrm{d}\pi(y) = \mathcal{K}_\pi \nabla \frac{\mathrm{d}\mu_t}{\mathrm{d}\pi}(x).$$

Thus, the continuous-dynamics of SVGD, as given in ($\mathrm{SVGD_d}$), can equivalently be expressed as

$$\partial_t \mu_t = \mathrm{div}\left(\mu_t \mathcal{K}_\pi \nabla \frac{\mathrm{d}\mu_t}{\mathrm{d}\pi}\right). \tag{SVGD}$$

To interpret this equation, we recall that the Wasserstein gradient of the chi-squared divergence $\chi^2(\cdot \,\|\, \pi)$ at $\mu$ is $2\nabla(\mathrm{d}\mu/\mathrm{d}\pi)$ (by (3)), so the gradient flow for the chi-squared divergence is

$$\partial_t \mu_t = 2\,\mathrm{div}\left(\mu_t \nabla \frac{\mathrm{d}\mu_t}{\mathrm{d}\pi}\right). \tag{CSF}$$

Comparing (SVGD) and (CSF), we see that (up to a factor of 2), SVGD can be understood as the flow obtained by replacing the gradient of the chi-squared divergence, $\nabla(\mathrm{d}\mu/\mathrm{d}\pi)$, by $\mathcal{K}_\pi \nabla(\mathrm{d}\mu/\mathrm{d}\pi)$.

Although ($\mathrm{SVGD_d}$) and (SVGD) are equivalent ways of expressing the same dynamics, the formulation of (SVGD) presents a significant advantage: it involves a kernel integral operator $\mathcal{K}_\pi$ that does not change with time and depends only on the target distribution $\pi$.

# 3  Chi-squared gradient flow

In this section, study the idealized case where $\mathcal{K}_\pi$ taken to be the identity operator. In this case, (SVGD) reduces to the gradient flow CSF. The existence, uniqueness, and regularity of this flow are studied in [OT11; OT13] and [AGS08, Theorem 11.2.1].

The rate of convergence of the gradient flow of the KL divergence is closely related to two functional inequalities: the Poincaré inequality controls the rate of exponential convergence in chi-squared divergence ([Pav14, Theorem 4.4], [Che+20]) while a log-Sobolev inequality characterizes the rate of exponential convergence of the KL divergence [BGL14, Theorem 5.2.1]. In this section, we show that these inequalities also guarantee exponential rates of convergence of CSF.

Recall that $\pi$ satisfies a *Poincaré inequality* with constant $C_P$ if

$$\operatorname{var}_\pi f \le C_P \, \mathbb{E}_\pi[\|\nabla f\|^2], \qquad \text{for all locally Lipschitz } f \in L^2(\pi), \tag{P}$$

while $\pi$ satisfies a *log-Sobolev inequality* with constant $C_{\mathsf{LSI}}$

$$\operatorname{ent}_\pi(f^2) := \mathbb{E}_\pi[f^2 \ln(f^2)] - \mathbb{E}_\pi[f^2] \ln \mathbb{E}_\pi[f^2] \le 2C_{\mathsf{LSI}} \, \mathbb{E}_\pi[\|\nabla f\|^2] \tag{LSI}$$

for all locally Lipschitz $f$ for which $\operatorname{ent}_\pi(f^2) < \infty$.

We briefly review some facts regarding the strength of these assumptions. It is standard that the log-Sobolev inequality is stronger than the Poincaré inequality: (LSI) implies (P) with constant $C_P \le C_{\mathsf{LSI}}$. In turn, if $\pi$ is $\alpha$-*strongly log-concave*, i.e. $\nabla^2 V \succeq \alpha I_d$, then it implies the validity of (LSI) with $C_{\mathsf{LSI}} \le 1/\alpha$, and thus a Poincaré inequality holds too. However, a Poincaré inequality is in general much weaker than strong log-concavity. For instance, if $\lambda_\pi^2$ denotes the largest eigenvalue of the covariance matrix of $\pi$, then it is currently known that $\pi$ satisfies a Poincaré inequality as soon as it is log-concave, with $C_P \le C(d)\lambda_\pi^2$, where $C(d)$ is a dimensional constant [Bob99; AB15; LV17], and the well-known *Kannan-Lovász-Simonovitz* (KLS) *conjecture* [KLS95] asserts that $C(d)$ does not actually depend on the dimension.

Our first result shows that a Poincaré inequality suffices to establish exponential decay of the KL divergence along CSF. In fact, we establish a remarkable property, which we call *strong uniform ergodicity*: under a Poincaré inequality, CSF forgets its initial distribution after a time of no more than $C_P/2$. Uniform ergodicity is central in the theory of Markov processes [MT09, Ch. 16] but is often limited to compact state spaces. Moreover, this theory largely focuses on total variation, so the distance from the initial distribution to the target distribution is trivially bounded by 1.

**Theorem 1.** *Assume that $\pi$ satisfies a Poincaré inequality* (P) *with constant $C_P > 0$ and let $(\mu_t)_{t \ge 0}$ denote the law of CSF. Assume that $\chi^2(\mu_0 \| \pi) < \infty$. Then,*

$$D_{\mathrm{KL}}(\mu_t \| \pi) \le D_{\mathrm{KL}}(\mu_0 \| \pi) \, e^{-\frac{2t}{C_P}}, \qquad \forall\, t \ge 0. \tag{6}$$

*In fact, a stronger convergence result holds:*

$$D_{\mathrm{KL}}(\mu_t \| \pi) \le \big(D_{\mathrm{KL}}(\mu_0 \| \pi) \wedge 2\big) \, e^{-\frac{2t}{C_P}}, \qquad \forall\, t \ge \frac{C_P}{2}. \tag{7}$$

*Proof.* Given the Wasserstein gradients (3) in Section 2.1, we get that $(\mu_t)_{t \ge 0}$ satisfies

$$\partial_t D_{\mathrm{KL}}(\mu_t \| \pi) = -2 \, \mathbb{E}_{\mu_t} \big\langle \nabla \ln \frac{\mathrm{d}\mu_t}{\mathrm{d}\pi}, \nabla \frac{\mathrm{d}\mu_t}{\mathrm{d}\pi} \big\rangle = -2 \, \mathbb{E}_\pi \big[\|\nabla \frac{\mathrm{d}\mu_t}{\mathrm{d}\pi}\|^2\big].$$

Applying the Poincaré inequality (P) with $f = \mathrm{d}\mu_t/\mathrm{d}\pi - 1$, we get

$$\partial_t D_{\mathrm{KL}}(\mu_t \| \pi) \le -\frac{2}{C_P} \chi^2(\mu_t \| \pi) \le -\frac{2}{C_P} D_{\mathrm{KL}}(\mu_t \| \pi),$$

where, in the last inequality, we use the fact that $D_{\mathrm{KL}}(\cdot \| \pi) \le \chi^2(\cdot \| \pi)$ (see [Tsy09, §2.4]). The bound (6) follows by applying Grönwall's inequality.

To prove (7), we use the stronger inequality $D_{\mathrm{KL}}(\cdot \| \pi) \le \ln[1 + \chi^2(\cdot \| \pi)]$ (see [Tsy09, §2.4]). Our differential inequality now reads:

$$\partial_t D_{\mathrm{KL}}(\mu_t \| \pi) \le -\frac{2}{C_P}\big(e^{D_{\mathrm{KL}}(\mu_t\|\pi)} - 1\big) \iff \partial_t \psi\big(D_{\mathrm{KL}}(\mu_t \| \pi)\big) \le -\frac{2}{C_P} \psi\big(D_{\mathrm{KL}}(\mu_t \| \pi)\big),$$

where $\psi(x) = 1 - e^{-x} \leq 1$. Grönwall's inequality now yields

$$\psi\big(D_{\mathrm{KL}}(\mu_t \,\|\, \pi)\big) \leq e^{-\frac{2t}{C_\mathsf{P}}} \psi\big(D_{\mathrm{KL}}(\mu_0 \,\|\, \pi)\big) \leq e^{-\frac{2t}{C_\mathsf{P}}}.$$

Note that $x \leq 2\psi(x)$ whenever $\psi(x) \leq 1/e$. Thus, if $t \geq C_\mathsf{P}/2$, we get $\psi\big(D_{\mathrm{KL}}(\mu_t \,\|\, \pi)\big) \leq e^{-1}$ so

$$D_{\mathrm{KL}}(\mu_t \,\|\, \pi) \leq 2\psi\big(D_{\mathrm{KL}}(\mu_t \,\|\, \pi)\big) \leq e^{-\frac{2t}{C_\mathsf{P}}},$$

which, together with (6), completes the proof of (7). $\qquad\square$

*Remark* 1. In [Che+20], it was observed that the chi-squared divergence decays exponentially fast along the gradient flow $(\mu_t)_{t \geq 0}$ for the KL divergence, provided that $\pi$ satisfies a Poincaré inequality. This observation is made precise and more general in [MMS09] where it is noted that the *gradient flow of a functional $\mathfrak{U}$ dissipates a different functional $\mathfrak{V}$ at the same rate that the gradient flow of $\mathfrak{V}$ dissipates the functional $\mathfrak{U}$*. A similar method is used to study the thin film equation in [CT02] and [Car11, §5].

Since we are studying the gradient flow of the chi-squared divergence, it is natural to ask whether CSF converges to $\pi$ in chi-squared divergence as well. In the next results, we show quantitative decay of the chi-squared divergence along the gradient flow under a Poincaré inequality (P), but we obtain only a polynomial rate of decay. However, if we additionally assume either that $\pi$ is log-concave or that it satisfies a log-Sobolev inequality (LSI), then we obtain exponential decay of the chi-squared divergence along CSF.

**Theorem 2.** *Suppose that $\pi$ satisfies a Poincaré inequality* (P). *Then, provided $\chi^2(\mu_0 \,\|\, \pi) < \infty$, the law $(\mu_t)_{t \geq 0}$ of* CSF *satisfies*

$$\chi^2(\mu_t \,\|\, \pi) \leq \chi^2(\mu_0 \,\|\, \pi) \wedge \big(\frac{9C_\mathsf{P}}{8t}\big)^2.$$

*If we further assume that $\pi$ is log-concave, then*

$$\chi^2(\mu_t \,\|\, \pi) \leq \chi^2(\mu_0 \,\|\, \pi)\, e^{-\frac{t}{2C_\mathsf{P}}}.$$

*Proof.* The proof is deferred to Appendix B. $\qquad\square$

Under the stronger assumption (LSI), we can show strong uniform ergodicity as in Theorem 1.

**Theorem 3.** *Assume that $\pi$ satisfies a log-Sobolev inequality* (LSI). *Let $(\mu_t)_{t \geq 0}$ denote the law of* CSF, *and assume that $\chi^2(\mu_0 \,\|\, \pi) < \infty$. Then, for all $t \geq 7C_{\mathsf{LSI}}$,*

$$\chi^2(\mu_t \,\|\, \pi) \leq \big(\chi^2(\mu_0 \,\|\, \pi) \wedge 2\big)\, e^{-\frac{t}{9C_{\mathsf{LSI}}}}.$$

*Proof.* The proof is deferred to Appendix B. $\qquad\square$

Convergence in chi-squared divergence was studied in recent works such as [CLL19; VW19; Che+20]. From standard comparisons between information divergences (see [Tsy09, §2.4]), it implies convergence in total variation distance, Hellinger distance, and KL divergence. Moreover, recent works have shown that the Poincaré inequality (P) yields transportation-cost inequalities which bound the 2-Wasserstein distance by powers of the chi-squared divergence [Din15; Led18; Che+20; Liu20], so we obtain convergence in the 2-Wasserstein distance as well. In particular, we mention that [Che+20] uses the chi-squared gradient flow (CSF) to prove a transportation-cost inequality.

## 4  Laplacian Adjusted Wasserstein Gradient Descent (LAWGD)

While the previous section leads to a better understanding of the convergence properties of SVGD in the case that $\mathcal{K}_\pi$ is the identity operator, it is still unclear how to choose the kernel $K$ to approach this idealized setup. For SVGD with a general kernel $K$, the calculation rules of Section 2.1 together with the method of the previous section yield the formula

$$\partial_t D_{\mathrm{KL}}(\mu_t \,\|\, \pi) = -\,\mathbb{E}_\pi\big\langle \nabla \frac{\mathrm{d}\mu_t}{\mathrm{d}\pi}, \mathcal{K}_\pi \nabla \frac{\mathrm{d}\mu_t}{\mathrm{d}\pi}\big\rangle,$$

for the dissipation of the KL divergence along SVGD. From this, a natural way to proceed is to seek an inequality of the form

$$\mathbb{E}_\pi\langle f, \mathcal{K}_\pi f\rangle \gtrsim \mathbb{E}_\pi[f^2], \qquad \text{for all locally Lipschitz } f \in L^2(\pi). \tag{8}$$

Applying this inequality to each coordinate of $\nabla(\mathrm{d}\mu_t/\mathrm{d}\pi)$ separately and using a Poincaré inequality would then allow us to conclude as in the proof of Theorem 1. The inequality (8) can be interpreted as a positive lower bound on the smallest eigenvalue of the operator $\mathcal{K}_\pi$. However, this approach is doomed to fail; under mild conditions on the kernel $K$, it is a standard fact that the eigenvalues of $\mathcal{K}_\pi$ form a sequence converging to $0$, so no such spectral gap can hold.[2]

This suggests that any approach which seeks to prove finite-time convergence results for SVGD in the spirit of Theorem 1 must exploit finer properties of the eigenspaces of the operator $\mathcal{K}_\pi$. Motivated by this observation, we develop a new algorithm called Laplacian Adjusted Wasserstein Gradient Descent (LAWGD) in which the kernel $K$ is chosen carefully so that $\mathcal{K}_\pi = \mathscr{L}^{-1}$ is the inverse of the generator of the Langevin diffusion that has $\pi$ as invariant measure.

More precisely, the starting point for our approach is the following integration-by-parts formula, which is a crucial component of the theory of Markov semigroups [BGL14]:

$$\mathbb{E}_\pi\langle \nabla f, \nabla g\rangle = \mathbb{E}_\pi[f\mathscr{L}g], \qquad \text{for all locally Lipschitz } f, g \in L^2(\pi), \tag{9}$$

where $\mathscr{L} := -\Delta + \langle\nabla V, \nabla\cdot\rangle$. The operator $\mathscr{L}$ is the (negative) generator of the standard Langevin diffusion with stationary distribution $\pi$ [Pav14, §4.5]. We refer readers to Appendix A for background on the spectral theory of $\mathscr{L}$.

In order to use (9), we replace the vector field $-\mathcal{K}_\pi\nabla(\mathrm{d}\mu_t/\mathrm{d}\pi)$ by the vector field $-\nabla\mathcal{K}_\pi(\mathrm{d}\mu_t/\mathrm{d}\pi)$. The new dynamics follow the evolution equation

$$\partial_t\mu_t = \mathrm{div}\big(\mu_t\nabla\mathcal{K}_\pi\frac{\mathrm{d}\mu_t}{\mathrm{d}\pi}\big). \tag{LAWGD}$$

The vector field in the above continuity equation may also be written

$$-\nabla\mathcal{K}_\pi\frac{\mathrm{d}\mu_t}{\mathrm{d}\pi}(x) = -\int \nabla_1 K(x,\cdot)\frac{\mathrm{d}\mu_t}{\mathrm{d}\pi}\,\mathrm{d}\pi = -\int \nabla_1 K(x,\cdot)\,\mathrm{d}\mu_t.$$

Replacing $\mu_t$ by an empirical average over particles and discretizing the process in time, we again obtain an implementable algorithm, which we give as Algorithm 1.

---
**Algorithm 1** LAWGD

1: **procedure** LAWGD($\mathsf{K}_\mathscr{L}, \mu_0$)
2:     draw $N$ particles $X_0^{[1]}, \ldots, X_0^{[N]} \overset{\text{i.i.d.}}{\sim} \mu_0$
3:     **for** $t = 1, \ldots, T-1$ **do**
4:         **for** $i = 1, \ldots, N$ **do**
5:             $X_{t+1}^{[i]} \leftarrow X_t^{[i]} - \frac{h}{N}\sum_{j=1}^N \nabla_1\mathsf{K}_\mathscr{L}(X_t^{[i]}, X_t^{[j]})$
6:         **end for**
7:     **end for**
8:     **return** $X_T^{[1]}, \ldots, X_T^{[N]}$
9: **end procedure**

---

A careful inspection of Algorithm 1 reveals that the update equation for the particles in Algorithm 1 does not involve the potential $V$ directly, unlike the SVGD algorithm (5); thus, the kernel for LAWGD must contain all the information about $V$.

Our choice for the kernel $K$ is guided by the following observation (based on (9)):

$$\partial_t D_{\mathrm{KL}}(\mu_t \parallel \pi) = -\mathbb{E}_\pi\big\langle\nabla\frac{\mathrm{d}\mu_t}{\mathrm{d}\pi}, \nabla\mathcal{K}_\pi\frac{\mathrm{d}\mu_t}{\mathrm{d}\pi}\big\rangle = -\mathbb{E}_\pi\big[\frac{\mathrm{d}\mu_t}{\mathrm{d}\pi}\mathscr{L}\mathcal{K}_\pi\frac{\mathrm{d}\mu_t}{\mathrm{d}\pi}\big].$$

As a result, we choose $K$ to ensure that $\mathcal{K}_\pi = \mathscr{L}^{-1}$. This choice yields

$$\partial_t D_{\mathrm{KL}}(\mu_t \parallel \pi) = -\mathbb{E}_\pi\big[\big(\tfrac{\mathrm{d}\mu_t}{\mathrm{d}\pi} - 1\big)^2\big] = -\chi^2(\mu_t \parallel \pi). \tag{10}$$

It remains to see which kernel $K$ implements $\mathcal{K}_\pi = \mathscr{L}^{-1}$. To that end, assume that $\mathscr{L}$ has a discrete spectrum and let $(\lambda_i, \phi_i)$, $i = 0, 1, 2, \dots$ be its eigenvalue-eigenfunction pairs where $\lambda_j$s are arranged in nondecreasing order. Assume further that $\lambda_1 > 0$ (which amounts to a Poincaré inequality; see Appendix A) and define the following *spectral kernel*:

$$\mathsf{K}_{\mathscr{L}}(x, y) = \sum_{i=1}^{\infty} \frac{\phi_i(x)\phi_i(y)}{\lambda_i} \tag{11}$$

We now show that this choice of kernel endows LAWGD with a remarkable property: it converges to the target distribution exponentially fast, with a rate which has no dependence on the Poincaré constant. Moreover, akin to CSF—see (7)—it also also exhibit strong uniform ergodicity.

**Theorem 4.** *Assume that $\mathscr{L}$ has a discrete spectrum and that $\pi$ satisfies a Poincaré inequality (P) with some finite constant. Let $(\mu_t)_{t \geq 0}$ be the law of LAWGD with the kernel described above. Then,*

$$D_{\mathrm{KL}}(\mu_t \parallel \pi) \leq \big(D_{\mathrm{KL}}(\mu_0 \parallel \pi) \wedge 2\big)\, e^{-t}, \qquad \forall\, t \geq 1.$$

*Proof.* In light of (10), the proof is identical to that of Theorem 1. $\qquad\square$

The convergence rate in Theorem 4 has no dependence on the target measure. This *scale-invariant convergence* also appears in [Che+20], where it is shown for the Newton-Langevin diffusion with a strictly log-concave target measure $\pi$. In Theorem 4, we obtain similar guarantees under the much weaker assumption of a Poincaré inequality; indeed, there are many examples of non-log-concave distributions which satisfy a Poincaré inequality [VW19].

## 5 Experiments

To implement Algorithm 1, we numerically approximate the kernel $K = \mathsf{K}_{\mathscr{L}}$ given in (11). When $\pi$ is the standard Gaussian distribution on $\mathbb{R}$, the eigendecomposition of the operator $\mathscr{L}$ in (9) is known explicitly in terms of the Hermite polynomials [BGL14, §2.7.1], and we approximate the kernel via a truncated sum: $\hat{K}(x, y) = \sum_{i=1}^{k} \lambda_i^{-1}\phi_i(x)\phi_i(y)$ (Figure 2) involving the smallest eigenvalues of $\mathscr{L}$.

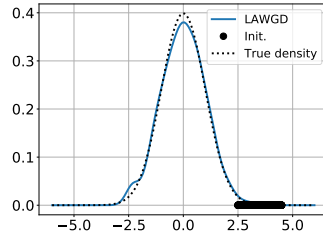

Figure 2: Samples from the standard Gaussian distribution generated by LAWGD, with kernel approximated by Hermite polynomials. For details, see Appendix C.

In the general case, we implement a basic finite difference (FD) method to approximate the eigenvalues and eigenfunctions of $\mathscr{L}$. We obtain better numerical results by first transforming the operator $\mathscr{L}$ into the Schrödinger operator $\mathscr{L}_{\mathsf{S}} := -\Delta + V_{\mathsf{S}}$, where $V_{\mathsf{S}} := \frac{1}{4}\|\nabla V\|^2 - \frac{1}{2}\Delta V$. If $\phi_{\mathsf{S}}$ is an eigenfunction of $\mathscr{L}_{\mathsf{S}}$ with eigenvalue $\lambda$ (normalized such that $\int \phi_{\mathsf{S}}^2 = 1$), then $\phi := e^{V/2}\phi_{\mathsf{S}}$ is an eigenfunction of $L$ also with eigenvalue $\lambda$ (and normalized such that $\int \phi^2\, \mathrm{d}\pi = 1$); see [BGL14, §1.15.7].

On a grid of points (with spacing $\varepsilon$), if we replace the Laplacian with the FD operator $\Delta_\varepsilon f(x) := \{f(x - \varepsilon) + f(x + \varepsilon) - 2f(x)\}/\varepsilon^2$ (in 1D), then the FD Schrödinger operator $\mathscr{L}_{\mathsf{S},\varepsilon} := -\Delta_\varepsilon + V_{\mathsf{S}}$ can be represented as a sparse matrix, and its eigenvalues and (unit) eigenvectors are found with standard linear algebra solvers.

When the potential $V$ is known only up to an additive constant, then the approximate eigenfunctions produced by this method are not normalized correctly; instead, they satisfy $\|\phi\|_{L^2(\pi)} = C$ for some constant $C$ (which is the same for each eigenfunction). In turn, this causes the kernel $K$ in LAWGD to be off by a multiplicative constant. For implementation purposes, however, this constant is absorbed in the step size of Algorithm 1. We also note that the eigenfunctions are differentiated using a FD approximation.

To demonstrate, we sample from a mixture of three Gaussians: $\frac{2}{5}\mathcal{N}(-3,1) + \frac{1}{5}\mathcal{N}(0,1) + \frac{2}{5}\mathcal{N}(4,2)$. We compare LAWGD with SVGD using the RBF kernel and median-based bandwidth as in [LW16]. We approximate the eigenfunctions and eigenvalues using a finite difference scheme, on 256 grid points evenly spaced between $-14$ and $14$. Constant step sizes for LAWGD and SVGD are tuned and the algorithms are run for 5000 iterations, and the samples are initialized to be uniform on $[1, 4]$. The results are displayed in Figure 3. All 256 discrete eigenfunctions and eigenvalues are used.

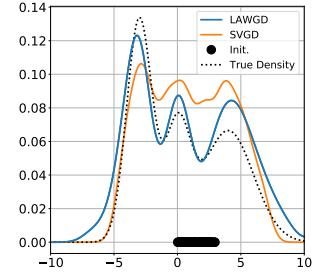

Figure 3: LAWGD and SVGD run with constant step size for a mixture of three Gaussians. Both kernel density estimators use the same bandwidth.

# 6   Open questions

We conclude this paper with some interesting open questions. The introduction of the chi-squared divergence as an objective function allows us to obtain both theoretical insights about SVGD and a new algorithm, LAWGD. This perspective opens the possibility of identifying other functionals defined over Wasserstein space and that yield gradient flows which are amenable to mathematical analysis and efficient computation. Towards this goal, an intriguing direction is to develop alternative methods, besides kernelization, which provide effective implementations of Wasserstein gradient flows. Finally, we note that LAWGD provides a hitherto unexplored connection between sampling and computing the spectral decomposition of the Schrödinger operator, the latter of which has been intensively studied in numerical PDEs. We hope our work further stimulates research at the intersection of these communities.

## Broader impact

The sampling algorithms designed in this paper have the potential to improve a wide variety of Bayesian methods and therefore have an indirect impact on various domains such as health and medicine where such methods are pervasive. Sampling algorithms are also used for the generation of automated spam messages, which have potentially negative effects on society. Since this paper is primarily focused on theory, these questions are not addressed here.

## Acknowledgments

Philippe Rigollet was supported by NSF awards IIS-1838071, DMS-1712596, DMS-TRIPODS-1740751. Sinho Chewi was supported by the Department of Defense (DoD) through the National Defense Science & Engineering Graduate Fellowship (NDSEG) Program. Thibaut Le Gouic was supported by ONR grant N00014-17-1-2147 and NSF IIS-1838071.

We thank the reviewers for very helpful suggestions regarding the presentation of the paper.

## Footnotes

[1]Throughout this paper, we call SVGD the generalization of the original method of [LW16; Liu17] that was introduced in [Wan+19].

[2]It is enough that $K$ is a symmetric kernel with $K \in L^2(\pi \otimes \pi)$, and that $\pi$ is not discrete (so that $L^2(\pi)$ is infinite-dimensional); see [BGL14, Appendix A.6].

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
