[Supplementary Material]

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

## A   Review of spectral theory

In this paper, we consider elliptic differential operators of the form $\mathscr{L} = -\Delta + \langle \nabla V, \nabla \cdot \rangle$, where $V$ is a continuously differentiable potential. In this section, we provide a brief review of the spectral theory of these operators, and we refer to [Eva10, §6.5] for a standard treatment.

The operator $\mathscr{L}$ (when suitably interpreted) is a linear operator defined on a domain $\mathscr{D} \subset L^2(\pi)$. For any locally Lipschitz function $f \in L^2(\pi)$, integration by parts shows that

$$\mathbb{E}_\pi[f\mathscr{L}f] = \mathbb{E}_\pi[\|\nabla f\|^2].$$

Therefore, $\mathscr{L}$ has a non-negative spectrum. Also, we have $\mathscr{L}1 = 0$, so that $0$ is always an eigenvalue of $\mathscr{L}$. We say that $\mathscr{L}$ has a *discrete spectrum* if it has a countable sequence of eigenvalues $0 = \lambda_0 \le \lambda_1 \le \lambda_2 \le \lambda_3 \le \cdots$ and corresponding eigenfunctions $(\phi_i)_{i=1}^\infty$ which form a basis of $\mathscr{D}$. The eigenfunctions can be chosen to be orthogonal and normalized such that $\|\phi_i\|_{L^2(\pi)} = 1$; we always assume this is the case. Then, $\mathscr{L}$ can be expressed as

$$\mathscr{L} = \sum_{i=1}^\infty \lambda_i \langle \phi_i, \cdot \rangle_{L^2(\pi)} \phi_i.$$

The operator $\mathscr{L}$ has a discrete spectrum under the following condition ([Fri34], [RS78, Theorem XIII.67], [BGL14, Corollary 4.10.9]):

$$V_{\mathsf{S}} \in L^1_{\mathrm{loc}}(\mathbb{R}^d), \qquad \inf V_{\mathsf{S}} > -\infty, \qquad \text{and} \qquad \lim_{\|x\|\to\infty} V_{\mathsf{S}}(x) = +\infty,$$

where $V_{\mathsf{S}} := -\Delta V + \frac{1}{2}\|\nabla V\|^2$. Moreover, under this condition we also have $\lambda_i \to \infty$ as $i \to +\infty$. For example, this condition is satisfied for $V(x) = \|x\|^\alpha$ for $\alpha > 1$, but not for $\alpha = 1$. In fact, for $\alpha = 1$, the spectrum of $\mathscr{L}$ is not discrete [BGL14, §4.1.1].

The *Poincaré inequality* (P) is interpreted as a *spectral gap inequality*, since it asserts that $\lambda_1 = 1/C_{\mathsf{P}} > 0$. Thus, under a Poincaré inequality, $\mathscr{L} : \mathscr{D} \cap \{f \in L^2(\pi) \mid \mathbb{E}_\pi f = 0\} \to L^2(\pi)$ is bijective. Moreover, if it has a discrete spectrum, its inverse satisfies

$$\mathscr{L}^{-1} = \sum_{i=1}^\infty \lambda_i^{-1} \langle \phi_i, \cdot \rangle_{L^2(\pi)} \phi_i.$$

# B Proofs of the convergence guarantees for the chi-squared flow

In this section, we give proofs of the convergence results we stated in Section 3.

*Proof of Theorem 2 (non-log-concave case).* According to [Che+20a, Proposition 1], the Poincaré inequality implies the following inequality for the chi-squared divergence:

$$\chi^2(\mu \parallel \pi)^{3/2} \leq \frac{9C_{\mathsf{P}}}{4} \, \mathbb{E}_\mu \big[ \| \nabla \frac{\mathrm{d}\mu}{\mathrm{d}\pi} \|^2 \big], \qquad \forall \mu \ll \pi. \tag{12}$$

Since the Wasserstein gradient of $\chi^2(\cdot \parallel \pi)$ at $\mu$ is given by $2\nabla(\mathrm{d}\mu/\mathrm{d}\pi)$ (see Section 2.1), it yields

$$\partial_t \chi^2(\mu_t \parallel \pi) = -4 \, \mathbb{E}_{\mu_t} \big[ \| \nabla \frac{\mathrm{d}\mu_t}{\mathrm{d}\pi} \|^2 \big] \leq -\frac{16}{9C_{\mathsf{P}}} \chi^2(\mu_t \parallel \pi)^{3/2}$$

Solving the above differential inequality yields

$$\chi^2(\mu_t \parallel \pi) \leq \frac{\chi^2(\mu_0 \parallel \pi)}{\{1 + 8t\sqrt{\chi^2(\mu_0 \parallel \pi)}/(9C_{\mathsf{P}})\}^2},$$

which implies the desired result. $\square$

We now prepare for the proof of exponentially fast convergence in chi-squared divergence for log-concave measures. The key to proving such results lies in differential inequalities of the form

$$\chi^2(\mu \parallel \pi) \leq C_{\mathsf{PL}} \, \mathbb{E}_\mu \big[ \| \nabla \frac{\mathrm{d}\mu}{\mathrm{d}\pi} \|^2 \big], \qquad \forall \mu \ll \pi, \tag{13}$$

which may be interpreted as a *Polyak-Łojasiewicz* (PL) *inequality* [KNS16] for the functional $\chi^2(\cdot \parallel \pi)$. PL inequalities are well-known in the optimization literature, and can be even used when the objective is not convex [Che+20b]. In contrast, the preceding proof uses the weaker inequality (12), which may be interpreted as a *Łojasiewicz inequality* [Loj63].

To see that a PL inequality readily yields exponential convergence, observe that

$$\partial_t \chi^2(\mu_t \parallel \pi) = -4 \, \mathbb{E}_{\mu_t} \big[ \| \nabla \frac{\mathrm{d}\mu_t}{\mathrm{d}\pi} \|^2 \big] \leq -\frac{4}{C_{\mathsf{PL}}} \chi^2(\mu_t \parallel \pi) \,.$$

Together with Grönwall's inequality, the differential inequality yields $\chi^2(\mu_t \parallel \pi) \leq \chi^2(\mu_0 \parallel \pi) \, e^{-\frac{4t}{C_{\mathsf{PL}}}}$.

In order to prove a PL inequality of the type (13), we require two ingredients. The first one is a transportation-cost inequality for the chi-squared divergence proven in [Liu20], building on the works [Din15; Led18]. It asserts that if $\pi$ satisfies a Poincaré inequality (P), then the following inequality holds:

$$W_2^2(\mu, \pi) \leq 2C_{\mathsf{P}} \chi^2(\mu \parallel \pi), \qquad \forall \mu \ll \pi. \tag{14}$$

For the second ingredient, we use an argument of [OV00] to show that if $\pi$ satisfies a chi-squared transportation-cost inequality such as (14), and in addition is log-concave, then it satisfies an inequality of the type (13). We remark that the converse statement, that is, if $\pi$ satisfies a PL inequality (13) then it satisfies an appropriate chi-squared transportation-cost inequality, was proven in [Che+20a] without the additional assumption of log-concavity. It implies that for log-concave distributions, the PL inequality (13) and the chi-squared transportation-cost inequality (14) are, in fact, equivalent.

**Theorem 5.** *Let $\pi$ be log-concave, and assume that for some $q \in (1, \infty)$ and a constant $\mathsf{C} > 0$,*

$$W_2^2(\mu, \pi) \leq \mathsf{C} \chi^2(\mu \parallel \pi)^{2/q}, \qquad \forall \, \mu \ll \pi.$$

*Then,*

$$\chi^2(\mu \parallel \pi)^{2/p} \leq 4\mathsf{C} \, \mathbb{E}_\mu \big[ \| \nabla \frac{\mathrm{d}\mu}{\mathrm{d}\pi} \|^2 \big], \qquad \forall \, \mu \ll \pi, \tag{15}$$

*where $p$ satisfies $1/p + 1/q = 1$.*

*Proof.* Following [OV00], let $T$ be the optimal transport map from $\mu$ to $\pi$. Since $\chi^2(\cdot \parallel \pi)$ is displacement convex [OT11; OT13] and has Wasserstein gradient $2\nabla(\mathrm{d}\mu/\mathrm{d}\pi)$ at $\mu$ (c.f. Section 2.1), the "above-tangent" formulation of displacement convexity ([Vil03, Proposition 5.29]) yields

$$0 = \chi^2(\pi \parallel \pi) \geq \chi^2(\mu \parallel \pi) + 2\,\mathbb{E}_\mu\big\langle\nabla\frac{\mathrm{d}\mu}{\mathrm{d}\pi}, T - \mathrm{id}\big\rangle \geq \chi^2(\mu \parallel \pi) - 2W_2(\mu,\pi)\sqrt{\mathbb{E}_\mu\big[\|\nabla\frac{\mathrm{d}\mu}{\mathrm{d}\pi}\|^2\big]},$$

where we used the Cauchy-Schwarz inequality for the last inequality. Rearranging the above display and using the transportation-cost inequality assumed in the statement of theorem, we get

$$\chi^2(\mu \parallel \pi) \leq 2W_2(\mu,\pi)\sqrt{\mathbb{E}_\mu\big[\|\nabla\frac{\mathrm{d}\mu}{\mathrm{d}\pi}\|^2\big]} \leq 2\sqrt{\mathsf{C}\,\mathbb{E}_\mu\big[\|\nabla\frac{\mathrm{d}\mu}{\mathrm{d}\pi}\|^2\big]}\,\chi^2(\mu \parallel \pi)^{1/q}.$$

The result follows by rearranging the terms. $\qquad\square$

*Proof of Theorem 2 (log-concave case).* From the transportation-cost inequality (14) and Theorem 5 with $p = q = 2$, we obtain

$$\chi^2(\mu \parallel \pi) \leq 8C_\mathsf{P}\,\mathbb{E}_\mu\big[\|\nabla\frac{\mathrm{d}\mu}{\mathrm{d}\pi}\|^2\big].$$

This PL inequality together with Grönwall's inequality readily yields the result. $\qquad\square$

We conclude this section with the proof of Theorem 3, which shows exponential convergence of CSF in chi-squared divergence under the assumption of a log-Sobolev inequality (LSI) (but without the assumption of log-concavity).

*Proof of Theorem 3.* We first claim that

$$\partial_t\chi^2(\mu_t \parallel \pi) \leq -\frac{4}{9C_{\mathsf{LSI}}}[\chi^2(\mu_t \parallel \pi) + 1]^{3/2}\ln[\chi^2(\mu_t \parallel \pi) + 1]. \tag{16}$$

Indeed, applying (LSI), we obtain

$$\partial_t\chi^2(\mu_t \parallel \pi) = -4\int\|\nabla\frac{\mathrm{d}\mu_t}{\mathrm{d}\pi}\|^2\,\mathrm{d}\mu_t = -\frac{16}{9}\int\|\nabla|\frac{\mathrm{d}\mu_t}{\mathrm{d}\pi}|^{3/2}\|^2\,\mathrm{d}\pi \leq -\frac{8}{9C_{\mathsf{LSI}}}\,\mathrm{ent}_\pi\big(|\frac{\mathrm{d}\mu_t}{\mathrm{d}\pi}|^3\big).$$

Next, the variational formula for the entropy gives

$$\mathrm{ent}_\pi f = \sup\{\mathbb{E}_\pi(fg) : g \text{ satisfies } \mathbb{E}_\pi\exp g = 1\},$$

see [Han16, Lemma 3.15] or [BLM13, Theorem 4.13]. Choosing $g = \ln(\mathrm{d}\mu_t/\mathrm{d}\pi)$ yields

$$\begin{aligned}
\mathrm{ent}_\pi\big(|\frac{\mathrm{d}\mu_t}{\mathrm{d}\pi}|^3\big) &\geq \mathbb{E}_\pi\big[|\frac{\mathrm{d}\mu_t}{\mathrm{d}\pi}|^3\ln\frac{\mathrm{d}\mu_t}{\mathrm{d}\pi}\big] = \frac{1}{3}\,\mathbb{E}_\pi\big[|\frac{\mathrm{d}\mu_t}{\mathrm{d}\pi}|^3\ln(|\frac{\mathrm{d}\mu_t}{\mathrm{d}\pi}|^3)\big] \\
&\geq \frac{1}{3}\,\mathbb{E}_\pi\big[|\frac{\mathrm{d}\mu_t}{\mathrm{d}\pi}|^3\big]\ln\mathbb{E}_\pi\big[|\frac{\mathrm{d}\mu_t}{\mathrm{d}\pi}|^3\big] \\
&\geq \frac{1}{2}\,\mathbb{E}_\pi\big[|\frac{\mathrm{d}\mu_t}{\mathrm{d}\pi}|^2\big]^{3/2}\ln\mathbb{E}_\pi\big[|\frac{\mathrm{d}\mu_t}{\mathrm{d}\pi}|^2\big] \\
&= \frac{1}{2}[\chi^2(\mu_t \parallel \pi) + 1]^{3/2}\ln[\chi^2(\mu_t \parallel \pi) + 1],
\end{aligned}$$

where in the second inequality, we used that $x \mapsto x\ln x$ is convex on $\mathbb{R}_+$ and in the third, we used that it increasing when $x \geq 1$ together with

$$\mathbb{E}_\pi\big[|\frac{\mathrm{d}\mu_t}{\mathrm{d}\pi}|^2\big] = 1 + \chi^2(\mu_t \parallel \pi) \geq 1.$$

This proves (16).

To simplify the inequality (16), we use the crude bounds

$$\ln[\chi^2(\mu_t \parallel \pi) + 1] \geq \begin{cases} 1, & \text{if } \chi^2(\mu_t \parallel \pi) \geq e - 1 \\ \chi^2(\mu_t \parallel \pi)/2, & \text{otherwise.} \end{cases}$$

It yields respectively

$$\partial_t \chi^2(\mu_t \,\|\, \pi) \leq -\frac{2}{9C_{\mathsf{LSI}}} \begin{cases} 2\chi^2(\mu_t \,\|\, \pi)^{3/2}, & \text{if } \chi^2(\mu_t \,\|\, \pi) \geq e - 1, \\ \chi^2(\mu_t \,\|\, \pi), & \text{otherwise.} \end{cases} \tag{17}$$

Solving the differential inequality in the first case yields

$$e - 1 \leq \chi^2(\mu_t \,\|\, \pi) \leq \left[ \frac{9C_{\mathsf{LSI}}\sqrt{\chi^2(\mu_0 \,\|\, \pi)}}{9C_{\mathsf{LSI}} + 2t\sqrt{\chi^2(\mu_0 \,\|\, \pi)}} \right]^2 \leq \left[ \frac{9C_{\mathsf{LSI}}}{2t} \right]^2,$$

so that in this first case, it must holds that

$$t \leq \frac{9C_{\mathsf{LSI}}}{2\sqrt{e-1}} < 3.5 C_{\mathsf{LSI}} =: t_0.$$

Therefore, if $t \geq t_0$, we are in the second case. In particular, $\chi^2(\mu_{t_0} \,\|\, \pi) \leq e - 1 \leq 2$ and integrating the differential inequality between $t_0$ and $t$ we get

$$\chi^2(\mu_t \,\|\, \pi) \leq \chi^2(\mu_{t_0} \,\|\, \pi) \, e^{-\frac{2(t-t_0)}{9C_{\mathsf{LSI}}}} \leq \left( \chi^2(\mu_0 \,\|\, \pi) \wedge 2 \right) e^{-\frac{2(t-t_0)}{9C_{\mathsf{LSI}}}},$$

where in the last inequality, we used the fact that $t \mapsto \chi^2(\mu_t \,\|\, \pi)$ is decreasing so that it also holds $\chi^2(\mu_{t_0} \,\|\, \pi) \leq \chi^2(\mu_0 \,\|\, \pi)$. In particular, taking $t \geq 2t_0 = 7C_{\mathsf{LSI}}$ yields the desired result. $\qquad\square$

## C Details for the experiments

We give additional details for the experiments presented in this paper. All methods were implemented in `Python`. Since the Schrödinger operator requires the Laplacian and gradient of the potential $V$, we employ automatic differentiation to avoid laborious calculations of these derivatives.

The *probabilists'* Hermite polynomials are well-known to be eigenfunctions of the 1D Ornstein-Uhlenbeck operator $\mathscr{L}$ given by $\mathscr{L}f(x) := -f''(x) + xf'(x)$, and they satisfy the recursive relationship $H_{n+1}(x) = xH_n(x) - nH_{n-1}(x)$, with $H_0(x) = 1$ and $H_1(x) = x$. It also holds that $H_n'(x) = nH_{n-1}(x)$. With these equations, it is easy to check that the eigenvalue corresponding to $H_n$ is $\lambda_n = n$. These are used as the eigenfunctions and eigenvalues in the standard normal example given in Figure 2. In the simulation, we use the first 150 Hermite polynomials. We run LAWGD for 2000 iterations with a constant step size, with initial points drawn uniformly from the interval $[2.5, 4.5]$.

In Figure 1, we display an example of sampling 50 particles from a mixture of two 2-dimensional Gaussian distributions given by $\pi = \frac{1}{2}\mathcal{N}((-1,-1)^\top, I_2) + \frac{1}{2}\mathcal{N}((1,1)^\top, I_2)$. To run this experiment, we use a 2-dimensional FD method, which approximates the Laplacian as

$$\Delta_\varepsilon f(x,y) := \frac{f(x-\varepsilon, y) + f(x+\varepsilon, y) + f(x, y-\varepsilon) + f(x, y+\varepsilon) - 4f(x)}{\varepsilon^2}.$$

We again use the Schrödinger operator for stability and use FD again to compute the gradients of the eigenfunctions. We use a $128 \times 128$ grid of evenly spaced $x$ and $y$ values between $-6$ and $6$. We calculate only the bottom 100 eigenvalues and eigenfunctions, since the other eigenfunctions incur additional computational cost without noticeably changing the result. Any negative eigenvalues (which arise from numerical errors) are discarded.

Additionally, we display the results from running SVGD with the RBF kernel and median-based bandwith on this example with a less favorable initialization. True samples from $\pi$ are displayed for comparison. Both LAWGD and SVGD are run for 20000 iterations with a constant step size. The samples from LAWGD tend to move very fast from their initial positions and then tend to settle into their final positions as seen in Figure 4. On the other hand, with constant step size, the samples of SVGD do not seem to converge, and one must use a decreasing step size scheme in order for the particles to stabilize. We also note that many of the samples generated by SVGD tend to blow up with a constant step size.

In Figure 5, we plot the particles of LAWGD and SVGD at iterations 100, 200, 1000, and 2000 to compare the speed of convergence.

Figure 4: Left: 50 particles and trajectories generated from $\frac{1}{2}\mathcal{N}((-1,-1)^\top, I_2) + \frac{1}{2}\mathcal{N}((1,1)^\top, I_2)$ with LAWGD. Middle: 50 particles and trajectories generated by SVGD. Right: true samples from the distribution.

Figure 5: Top: LAWGD after 100, 200, 1000, and 2000 iterations. Bottom: SVGD after 100, 200, 1000, and 2000 iterations.

# D   Broader impact

The sampling algorithms designed in this paper have the potential to improve a wide variety of Bayesian methods and therefore have an indirect impact on various domains such as health and medicine where such methods are pervasive. Sampling algorithms are also used for the generation of automated spam messages, which have potentially negative effects on society. Since this paper is primarily focused on theory, these questions are not addressed here.

# E   Acknowledgments

Philippe Rigollet was supported by NSF awards IIS-1838071, DMS-1712596, DMS-TRIPODS-1740751. Sinho Chewi was supported by the Department of Defense (DoD) through the National Defense Science & Engineering Graduate Fellowship (NDSEG) Program. Thibaut Le Gouic was supported by ONR grant N00014-17-1-2147 and NSF IIS-1838071.

We thank the reviewers for very helpful suggestions regarding the presentation of the paper.

## Footnotes

[1]Throughout this paper, we call SVGD the generalization of the original method of [LW16; Liu17] that was introduced in [Wan+19].

[2]It is enough that $K$ is a symmetric kernel with $K \in L^2(\pi \otimes \pi)$, and that $\pi$ is not discrete (so that $L^2(\pi)$ is infinite-dimensional); see [BGL14, Appendix A.6].