[Reviews · NeurIPS 2020]

Review 1

Summary and Contributions: In this paper, the authors give us a new perspective on Stein Variational Gradient Descent (SVGD). Instead of Kullback-Leibler divergence, they view SVGD as the (kernelized) gradient flow of the chi-squared divergence, which exhibits a strong form of uniform exponential ergodicity under conditions as weak as a Poincare inequality. Based on theoretical analysis, they also propose an alternative to SVGD, called Laplacian Adjusted Wasserstein Gradient Descent (LAWGD).

Strengths: 1. They provide detailed theoretical analyses on SVGD. It seems to be reasonable. 2. Based on these theoretical analyses, they provide a new algorithm called LAWGD.

Weaknesses: 1. The most concern about this work is the experiments. Although this paper is equipped with powerful theoretical analysis, its experiments about LAWGD is weak. I am not very familiar with SVGD but I think the experiment is not enough for NeurIPS. For example, in [1], authors apply SVGD on Bayesian Logistic Regression and Bayesian Neural Network with different tasks. Compared with [1], this work only illustrates the advantage of LAWGD on simulated data such as mixture of Gaussian distribution. If the authors can provide more experimental results on LAWGD, I think I may improve my score. 2. The theoretical analyses take up the main space of the paper. For many of readers, they may not very familiar with such detailed analysis. Most of them want to know about the difference of this work with existing ones. However, in the paper, such discussions are few, for example, the related works is a little weak. Actually, the first reason is the main reason that I give the score 5. The second reason is my personal suggestion about how to make a theoretical paper more easier to understand for most of readers. [1] Stein Variational Gradient Descent: A General Purpose Bayesian Inference Algorithm

Correctness: Actually, I am not very familiar with this field. I try my best to understand their method and proofs. I think it is correct based on my current knowledge.

Clarity: The paper is well written. However, as my personal suggestion, I think the authors can give more discussions on the difference or novelty of their work.

Relation to Prior Work: Not very clear.

Reproducibility: Yes

Additional Feedback:


Review 2

Summary and Contributions: The paper makes the following contributions: 1) Interpretation (up to a constant factor of 2) of SVGD as (kernelized) gradient flow of the Chi-squared divergence, called as CSF 2) Establishing exponential ergodicity of CSF (continuous case) with respect to the KL metric and Chi-squared divergence metric, under certain Poincare condition (or LSI) on the target. 3) Propose the use of Laplacian based kernel for SVGD and establish scale-invariant ergodicity results under certain assumptions. =====updates after response===== Line 11 in response mentions kernel selection is an issue. Indeed this is an issue with any kernel method (from SVM to MMD to SVGD) and it has been addressed in various ways. If one were critical, there is still no "nice" way to pick a kernel. Indeed as mentioned in Line 16 and 17 , a single integral operator depending on target \pi is good (in a way it is also along expected lines - for example in MMD context something similar leads to optimality properties). However I tend to not agree 100% with lines 27-28 that "solving high-dimensional PDEs is precisely the target of intensive research in modern numerical PDE" which is my main concern with the practical applicability of the proposed work. There is no "concrete" progress in this direction to the best of the reviewer's knowledge despite several ad-hoc approaches recently. However, I also recognize making concrete progress in high-dim numerical pde is not the main aim of this paper. Given this, I am still skeptical of the practicality of the proposed approach (there is a good possibility it will remain difficult to compute for various \pi and dimension combinations -- I would be delighted to be proven wrong~!! This is the case for many practical kernel selection methods unfortunately). Hence, I retain my score.

Strengths: 1) The interpretation of SVGD (up to a constant) as Chi-squared flow is interesting. 2) The proposal of Laplacian based kernel for SVGD is interesting.

Weaknesses: 1) The main issue with the paper is the lack of results or algorithms in the discrete setting. In particular, how to compute the Laplacian-based kernel efficiently is completely left as open question with connection to numerical PDE literature. It could be argued that the numerical techniques in the PDE literature are not scalable and hence this severely limits the applicability of the proposed method. Do the authors have more to comment on this aspect ? 2) The establishment of exponential ergodicity conditions are more or less standard given the flow interpretation. Hence, the main contribution of the paper is in terms of an "idea" or a "method" with some potential.

Correctness: To my understanding the claims and the methods are correct.

Clarity: yes -- very easy to read and understand

Relation to Prior Work: yes.

Reproducibility: Yes

Additional Feedback:


Review 3

Summary and Contributions: Stein variational gradient descent (SVGD) is a particle-based sampling algorithm that have attracted much attention recently. While SVGD has typically been viewed as the kernelized gradient flow of the KL-divergence, this paper offers a new perspective by interpreting it as the kernelized gradient flow of the Chi-squared divergence. Adopting this perspective, the paper proves uniform ergodicity under a Poincare inequality. Motivated by finding an optimal kernel, the paper further proposes an alternative sampling algorithm which they term Laplacian Adjusted Wasserstein Gradient Flow (LAWGD) that exhibits stronger convergence guarantees.

Strengths: SVGD and its analysis from the perspective of gradient-flows have attracted much attention recently, and I believe this paper will be a refreshing addition to the literature. The paper is clearly written, and the derivation of SVGD as the kernelized gradient flow of the Chi-squared divergence is simple and rather elegant. It’s nice to see the advantage of this interpretation over SVGD_p, namely that the kernel integral operator depends only on the target distribution and is stationary through time. While the proofs are not very complex, the results on strong uniform ergodicity under Poincare inequality is quite appealing. I also find the motivation and derivation of the LAWGD algorithm from the generator of the Langevin diffusion insightful.

Weaknesses: An obvious limitation of the current work is that the current implementation of the LAWGD algorithm is merely a proof-of-concept, and much more efficient numerical solvers should be used in place of the current finite-difference scheme to make the algorithm actually practical. However, this is fine with me as the paper’s primary focus is theoretical, and would probably inspire follow-up research on developing more practical algorithms. I should also note that while the theoretical results look appealing to me, I am not sufficiently familiar with the recent developments in the learning theory community to comment on the novelty and innovativeness of the analysis techniques.

Correctness: To the best of my knowledge, yes; but I have not checked the proofs step-by-step.

Clarity: Yes, the paper is clearly written and a pleasure to read.

Relation to Prior Work: I believe the paper has addressed prior work appropriately, but I may be missing very recent developments in the learning theory community.

Reproducibility: Yes

Additional Feedback: I thank the authors for their response, and I also think Reviewers 2 and 4 raised valid points regarding the lack of discrete-time results. On the other hand, regarding the lack of extensive or practical experiments/applications in this paper, while this is definitely the case, I believe that a simple proof-of-concept is sufficient for a paper whose focus is theoretical, and that the present paper would inspire future research that develop more practical algorithms. In this regard, I'm happy to raise my overall score from 7 to 8. ================================================== I have some more specific comments/questions: - While the derivation of the Chi-squared divergence perspective does not exhibit a direct analogy to the case of the KL-divergence, could similar interpretations be established for other f-divergences? - A well-known problem with SVGD is that in high dimensions, the particles tend to collapse around some modes of the target distribution and fail to spread out. Theoretically speaking, would LAWGD also suffer from this issue using the same e.g., RBF kernel, or would one expect LAWGD to more fully explore the parameter-space?
 - As a consequence of Equations (10) and (11), it’s somewhat interesting that to approximate K_\Lcal one would be interested in the bottom of the spectrum of \Lcal rather than the largest eigenvalues. Would this give rise to any practical (numerical) implications in discretizing the algorithm? - It’s quite interesting to see the connection between particle-based sampling and the spectrum of the Schrodinger operator. More generally, could LAWGD potentially be derived from the Schrodinger operator rather than the generator for the Langevin diffusion instead? - Minor: In Equation (8), \gtrsim is not properly defined.


Review 4

Summary and Contributions: This paper first analyzes the popular SVGD algorithm as a kernelized gradient flow of the chi-squared divergence and further proposes Laplacian adjusted Wasserstein gradient descent, another particle-based variational inference method. Given some assumptions on the derivatives of the target distribution, the new method can have a faster convergence rate.

Strengths: [+] Detailed discussions about the background and related work, e.g. Wasserstein SVGD as gradient flow, accelerated version of SVGD. [+] Well-organized methodlogy part and show the advance of the proposed algorithm clearly.

Weaknesses: [-] Following the analysis of SVGD with KL divergence, the authors analyzed the dynamics for the chi-square divergence. I would like to see some further analysis of the discretization of the gradient flow, which is more difficult but more meaningful. [-] Compared to [1] and its variants, the difference is convergence rate in Theorem 4, which should be highlighted. [-] (Main) My main concern is about the experiment part. This paper proposes a new algorithm, and I think the experiments should empirically show the advances of the algorithm. (Baselines) For baselines, besides SVGD, some recent works discussed in section 2 should be compared, e.g. [1], [3]. These algorithms also claim they can accelerate SVGD empirically and can improve the performance. (Time complexity) The proposed Laplacian adjusted Wasserstein gradient descent needs to calculate the eigendecomposition. Therefore, I would like to see some running time results for real-world applications, e.g. Bayesian neural network, generative model [2]. (Practical) I would like to see some experiments (e.g. Bayesian neural network, generative model, etc.) except the toy cases (e.g. Gaussian mixture), using Laplacian adjusted Wasserstein gradient descent. Otherwise, I would wonder whether this is a practical algorithm. [1] A Stein variational Newton method [2] Kernel Stein Generative Modeling [3] Stein variational gradient descent with matrix-valued kernels

Correctness: The claims and methods are correct.

Clarity: The paper is well-written.

Relation to Prior Work: It clearly discussed how this work differs from the previous contribution.

Reproducibility: Yes

Additional Feedback:

[Author Response · NeurIPS 2020]

We thank the reviewers for their thoughtful feedback. It seems we have not sufficiently clarified the context and scope
of our contributions, so we begin with general comments in this direction. We will update the introduction of our
paper accordingly. We thank the reviewers for bringing this to our attention.

**Clarifying the context and scope of our contributions**. SVGD is markedly different from sampling methods based
on diffusions, such as Langevin or HMC, in that it constructs a sequence of *deterministic* mappings whose composition
approximately pushes forward an initial distribution to the target distribution. The discovery of its interpretation as
a gradient flow of the KL divergence in the Wasserstein space of probability measures, as well as a connection to
the rich mathematical theory of interacting particle systems, has led to great interest among researchers. (As noted
by **R2**, a gradient flow interpretation lends itself easily to convergence analysis.) However, despite fervent activity,
the prevailing perspective has not provided sufficient theoretical understanding for SVGD to overcome its known
problems, such as mode collapse (**R3**) or lack of guidance on how to choose an appropriate kernel, and consequently
diffusion-based algorithms remain the dominant choice for applications.

In this work, we provide a new and stronger theoretical footing for the development of such deterministic mappings.
In particular, it allows us to derive improved convergence guarantees for SVGD (including strong uniform ergodicity,
a property which is not standard in the literature; **R2**). Additionally, by introducing a simpler conceptual framework
in which the properties of a single integral operator, $\mathcal{K}_\pi$, governs the performance of SVGD, our work is the first to
demonstrate that the interplay of the kernel and the target distribution is crucial for designing SVGD-like algorithms.
As a proof of concept, our proposed algorithm uses a kernel which is carefully designed based on the target distribu-
tion. We believe our study will stimulate further work on the design of sampling algorithms, based on deterministic
pushforward mappings, which may eventually see the same widespread application as diffusion-based algorithms.

In fact, we do not advocate for LAWGD as the definitive solution of the sampling problem, but rather as the start of a
family of interacting particle systems with an interacting potential that depends strongly and non-trivially on the target
distribution and furthermore comes with strong theoretical guarantees.

**On the role of numerical PDEs**. LAWGD establishes a firm bridge between the fields of sampling and numerical
PDEs, whereby the main computational bottleneck of our algorithm is the inversion of a differential operator. Although
our naïve implementation of LAWGD is not scalable to high dimension (as rightly pointed out by **R2** and **R4**), the
problem of efficiently solving high-dimensional PDEs is precisely the target of intensive research in modern numerical
PDEs, culminating in a wide variety of methods ranging from ad-hoc but effective neural network approaches to more
principled solvers. We view this bridge as one of our core contributions and we hope that attracting the attention of
numerical PDE researchers will yield fruitful collaborations for both fields.

**Addressing specific comments**.

• **R1** and **R4** write that our experiments are not extensive enough. This is an excellent point and we fully agree
that more in-depth experiments are required to evaluate the practicality of LAWGD. As mentioned above,
however, our goal is not to establish the supremacy of LAWGD by testing it on a battery of challenging high-
dimensional instances, but rather to demonstrate that, unlike SVGD, LAWGD has both strong theoretical
guarantees and good numerical performance (recall that for non-trivial kernels there is currently no quantita-
tive convergence analysis of SVGD under verifiable assumptions and that even our simple experiments on the
mixture of two Gaussians demonstrate a failure of SVGD). This is an indication that our novel perspective
could be the correct one to further advance the state-of-the-art for sampling via deterministic mappings.

• **R2** and **R4** note that we do not provide analysis in discrete time. Although discrete time analysis is common
for the more established class of diffusion-based sampling algorithms, the understanding of SVGD-like algo-
rithms is still nascent. Following the trend of recent work such as Duncan et al., we work in continuous time
in order to develop conceptual understanding regarding the impact of the choice of kernel.

• **R3** raises a number of interesting questions, and we address some here. Just as we have established SVGD as
a kernelized gradient flow of the chi-squared divergence, it would indeed be interesting to consider gradient
flows of other functionals, such as f-divergences, and to use kernels or develop other approaches to implement
them; we mention these directions in our open questions.

For LAWGD, the choice of kernel is fully determined by the target distribution, and in principle there is no
risk of the kernel being mismatched to the target distribution (as in the case of the observed failure of SVGD
with RBF kernels). In practice, however, more numerical experiments are necessary to determine if LAWGD
suffers from similar problems as SVGD in high dimension.

We also note your thought-provoking question about a more direct connection between LAWGD and the
Schrödinger operator. While we cannot see an obvious connection in the context of sampling (for which $\mathscr{L}$
is more natural), it is possible that it becomes a more central object in particle methods for PDEs without a
limiting distribution.

[Meta-Review · NeurIPS 2020]

The reviewers had some concerns about the empirical evaluation and the lack of discrete-time results, but agree that this paper would be a useful addition to NeurIPS. Please see the reviews (and your response) for ways to improve the final manuscript (especially in terms of clarifying the context and scope of this work).